# Inherent spatiotemporal uncertainty of renewable power in China

Jianxiao Wang ©[1,2,13], Liudong Chen[3,13], Zhenfei Tan ©[4], Ershun Du[5], Nian Liu[6], Jing Ma[6], Mingyang Sun[7], Canbing Li[4], Jie Song[8,1,2], Xi Lu ©[9,10] ✉, Chin-Woo Tan[11] ✉ & Guannan He ©[1,8,12] ✉

Solar and wind resources are vital for the sustainable energy transition. Although renewable potentials have been widely assessed in existing literature, few studies have examined the statistical characteristics of the inherent renewable uncertainties arising from natural randomness, which is inevitable in stochastic-aware research and applications. Here we develop a rule-of-thumb statistical learning model for wind and solar power prediction and generate a year-long dataset of hourly prediction errors of 30 provinces in China. We reveal diversified spatiotemporal distribution patterns of prediction errors, indicating that over 60% of wind prediction errors and 50% of solar prediction errors arise from scenarios with high utilization rates. The first-order difference and peak ratio of generation series are two primary indicators explaining the uncertainty distribution. Additionally, we analyze the seasonal distributions of the provincial prediction errors that reveal a consistent law in China. Finally, policies including incentive improvements and interprovincial scheduling are suggested.

To realize China's carbon neutrality goal proposed in 2020[1], the installed capacity of renewable energy resources should be significantly increased. As China mentioned in the 2020 Climate Ambition Summit, the installation of wind and solar energy should reach no less than 1.2 Terawatt (TW) in 2030, almost 3 times more than that in 2019[2], becoming the dominant electricity generation resource. However, due to the salient intermittency and volatility, wind and solar energy operation and modeling face the critical challenges of a high degree of uncertainty, which must be considered in energy research[3–5].

Various studies have investigated the generalized spatial and temporal characteristics of renewable energy resources in regional areas and compiled standardized test datasets, including statistical analysis studies of current wind and solar resources[6–10] and important impact factors of renewable energy generation[11], current wind and solar energy resource estimation studies using meteorological data and prediction methods[12–14], and future wind and solar energy resource assessment studies based on wind speed and solar irradiation data[15–19]. However, renewable energy resources rely on weather conditions and thus are highly unstable, posing great challenges to accurate and

[1]National Engineering Laboratory for Big Data Analysis and Applications, Peking University, Beijing 100871, China. [2]Peking University Ordos Research Institute of Energy, Ordos 017000, China. [3]Department of Earth and Environmental Engineering, Columbia University, New York, NY 10027, USA. [4]Key Laboratory of Control of Power Transmission and Conversion (Ministry of Education), Shanghai Jiao Tong University, Shanghai 200240, China. [5]Low-Carbon Energy Laboratory, Tsinghua University, Beijing 100084, China. [6]State Key Laboratory of Alternate Electrical Power System with Renewable Energy Sources, School of Electrical and Electronic Engineering, North China Electric Power University, Beijing 102206, China. [7]College of Control Science and Engineering, Zhejiang University, Hangzhou 310058, China. [8]Department of Industrial Engineering and Management, College of Engineering, Peking University, Beijing 100871, China. [9]School of Environment and State Key Joint Laboratory of Environment Simulation and Pollution Control, Tsinghua University, Beijing 100084, China. [10]Institute for Carbon Neutrality, Tsinghua University, Beijing 100084, China. [11]Department of Civil and Environmental Engineering, Stanford University, Palo Alto, CA 94305, USA. [12]Institute of Carbon Neutrality, Peking University, Beijing 100871, China. [13]These authors contributed equally: Jianxiao Wang, Liudong Chen. ✉e-mail: xilu@tsinghua.edu.cn; tancw@stanford.edu; gnhe@pku.edu.cn

reliable prediction. Some studies have examined the uncertainty of solar and wind power equipped with energy storage to assess their potential to meet future electricity demand[20]. Prediction methods such as linear regression models and eXtreme Gradient Boosting have been utilized to forecast the uncertainty of wind and solar generation in specific regional areas, considering seasonal or yearly analyses[21,22]. However, limited research has focused on analyzing the spatio-temporal uncertainty distributions of renewable energy[23,24]. There are research gaps in terms of error analysis benchmarks that consider long-term, high-granularity, and nationwide scales of wind and solar output prediction, particularly within the context of China.

Error-analysis benchmarks for wind and solar output forecasting are of great value in academic research and industry. First, a prediction error database of the wind and solar output should be obtained via benchmark prediction methods, e.g., neural network-based[25], data mining[26], and regression methods[27]. Second, a wide variety of studies, e.g., power system planning and operation[28–31], energy scheduling[32–34], and market operation and mechanism design studies[35,36], must consider the intermittency and volatility of renewable energy resources via robust optimization[37,38], stochastic programming[39,40], and statistical analysis methods[41,42]. Third, the prediction error of renewable power determines the revenue risk of power generation companies, especially in markets with deviation punishment. In this regard, prediction error analysis can provide an important reference for the decision-making of intermittent renewables.

The motivation of this work is to develop a year-long error-analysis benchmark for hourly wind and solar generation forecasting in 30 provinces of China, which is expected to constitute a valuable resource and toolkit for market operators or planners. To this end, we use a one-year standard dataset including hourly wind and solar output data for 30 provinces of China[11]. Here, we establish a rule-of-thumb prediction model to conduct hourly predictions of the wind and solar output in a rolling fashion and to obtain basic prediction datasets. The results reveal the nationwide spatial distribution of the wind and solar energy uncertainty through the prediction error. The first-order difference and peak ratio of output data are determined as primary factors of the prediction error. To further analyze provincial forecasting characteristics, we provide the provincial probability distribution function (PDF) of prediction errors and distribution regularities, the influence of power generation intervals on prediction in each province, and the temporal features of uncertainty via seasonal analysis.

## Results
### Nationwide analysis of the uncertainty of wind and solar generation
We obtain an error-analysis benchmark for the forecasting of hourly wind and solar output potential in 30 provinces of China in 2016 using the autoregressive integrated moving average (ARIMA) model based on installation and hourly generation data retrieved from our previous study[11]. The spatial distributions of the wind and solar uncertainty across China are analyzed through the prediction error, as shown in Fig. 1a, b, respectively, excluding Taiwan, Hong Kong, and Macau, as well as wind energy in Tibet and solar energy in Chongqing (unsuitable for wind/solar energy construction[10] or data limitations). The prediction error is calculated as the predicted value minus the actual value (please refer to Methods). The wind prediction error ranges from 2.1 to 13.6%, with the largest error in Tianjin (TJ) and the smallest error in Yunnan (YN). The overall prediction error of solar energy is smaller than that of wind energy, ranging from 3.9 to 10.0%, and the largest provincial prediction error is observed in Shanghai (SH), while the smallest provincial prediction error comes from Xinjiang (XJ). Detailed error analysis of wind and solar power for each province is shown in Supplementary Figs. 1–3, respectively. We divide the 30 provinces into four groups according to the wind prediction error: (i) >9%, (ii) 7–9%, (iii) 5–7%, and (iv) <5%. Four groups can also be distinguished in terms

of solar energy according to the prediction error: (i) >8%, (ii) 7–8%, (iii) 6–7%, and (iv) <6%. The details of each group are provided in the Supplementary Information (SI).

The results demonstrate that, except for Southwest China, the wind prediction error in the other regions is relatively large, especially large in the eastern area, i.e., Shandong (SD), SH, Jiangsu (JS), Anhui (AH), and Henan (HA), and Northern area including Beijing (BJ), TJ, Liaoning (LN), Jilin (JL), Shanxi (SX), and Hebei (HE), ranging from 8.0 to 11.3% and 5.3 to 13.6%, respectively. These two areas account for 25.0% and 27.9%, respectively, of the total prediction error in China. Regarding solar energy, the prediction error is concentrated in the areas of Central China covering Ningxia (NX), Shaanxi (SN), Hubei (HB), Jiangxi (JX), and Hunan (HN), North China, and East China, ranging from 6.2 to 9.0%, 7.2 to 9.3%, and 6.8 to 10.0%, respectively, accounting for 17.5%, 25.0%, and 19.1%, respectively, of the total prediction error in China.

We compare the prediction errors of various methods, including random forest (RF), recurrent neural network (RNN), fully-connected neural network (FCNN), and support vector machine (SVM), for predicting nationwide renewable energy output. The results are presented in Fig. 1c and Supplementary Table 1. Our observations indicate that although each method demonstrates varying prediction error distributions across different provinces, the overall nationwide prediction errors remain similar among all methods, ranging from 6 to 9%. Further details can be found in the SI. Notably, ARIMA and RNN exhibit similar prediction errors and outperform other methods, benefiting from their inherent ability to effectively handle time series data. In the following part of this paper, we focus on the prediction error with the ARIMA model as a benchmark method.

Moreover, we examine the impact of the prediction time scale on the distribution of nationwide prediction errors for both wind and solar energy, as illustrated in Fig. 1d. We observe that prediction error increases with the prediction time scale, with a 2-h prediction resulting in a 3.40% error for solar and a 2.83% error for wind, a 6-h prediction resulting in a 6.14% error for solar and a 6% error for wind, and a 24-h prediction resulting in a 9.25% error for solar and a 10.86% error for wind. A detailed analysis of each hour's prediction error reveals that the error mainly originates from the ending periods, e.g., during 5–6 h for the 6-h ahead predictions and during 15–24 h for the 24-h ahead predictions.

### Key factors affecting prediction errors
Two statistical indicators are proposed to explore the factors impacting prediction errors. Due to the irregular distribution of the wind output and the daily periodicity of the solar output, we use hourly and daily output data to analyze the wind and solar prediction errors, respectively (Methods and Supplementary Fig. 4). We use the coefficient of determination (CoD) $R^2$, which measures the linear correlation, to quantify the relationship between the prediction error and various factors. The installed capacity is independent of the prediction error, with $R^2 = 0.002$ for wind energy (Fig. 2a) and $R^2 = 0.076$ for solar energy (Fig. 2b). In addition, the power generation reflected by the bubble size exhibited no correlation with the prediction error (Fig. 2a, b).

As shown in Fig. 2c, d, the results indicate that the first-order difference is a major influencing factor of the prediction error, which comprises a series of changes from one period to the next. The relationship between the prediction error and first-order difference is approximately linear. Regarding wind power, the relationship between the prediction error and hourly first-order difference yields $R^2 = 0.988$ (Fig. 2c), while the daily first-order difference does not impact the wind prediction error (please refer to the bubble size in Fig. 2c). Regarding solar power, the CoD between the prediction error and the daily first-order difference is $R^2 = 0.676$ (Fig. 2d). The hourly first-order difference, however, could not reflect the prediction error, as indicated by

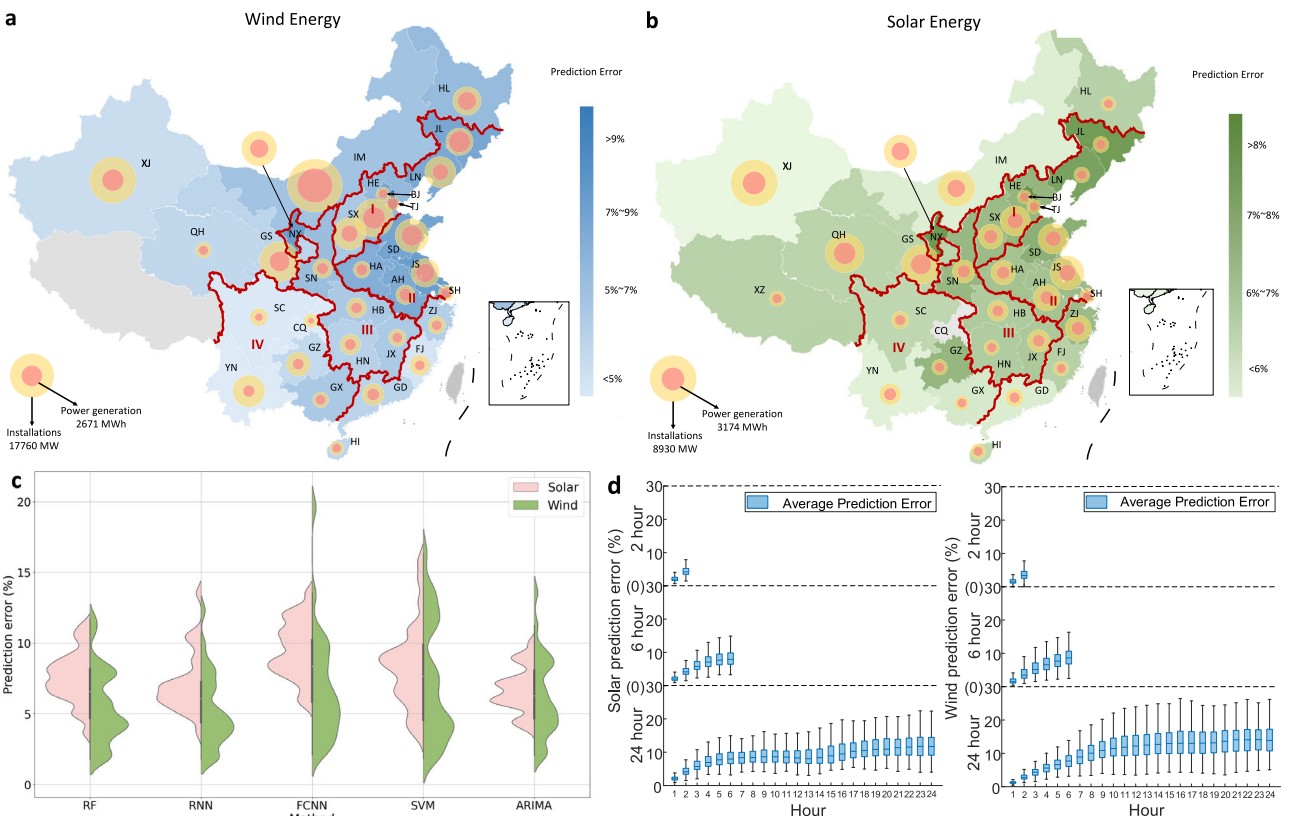

**Fig. 1 | Spatial distributions of wind and solar power prediction errors and the impacts of different methods and time scales. a** Wind energy. **b** Solar energy. The larger bubbles indicate the provincial wind and solar energy installations, and the smaller ones indicate the average wind and solar energy generation (8760 h) by province. The provinces are divided into four groups according to the provincial prediction error (average value of 8760 h) and marked with four gradient colors. The thick red line marks the boundaries of the four areas of China, I. North China, II. East China, III. Central China, and IV. Southwest China. Individual provinces are indicated with lighter white lines. MW Megawatt, BJ Beijing, TJ Tianjin, HE Hebei, SX Shanxi, IM Inner Mongolia, LN Liaoning, JL Jilin, HL Heilongjiang, SH Shanghai, JS Jiangsu, ZJ Zhejiang, AH Anhui, FJ Fujian, JX Jiangxi, SD Shandong, HA Henan, HB Hubei, HN Hunan, GD Guangdong, GX Guangxi, HI Hainan, CQ Chongqing, XZ Tibet, SC Sichuan, GZ Guizhou; YN Yunnan, SN Shaanxi, GS Gansu, QH Qinghai, NX Ningxia XJ Xinjiang. **c** Prediction error distribution across 30 provinces obtained by different methods. The smoothed curve in the left and right parts represents the prediction error density function across 30 provinces of solar and wind energy, respectively. The short black line in the middle of each shape is the median value of the data distribution, which visualizes the central tendency of the data distribution of each method. The algorithm used to fit the density function is Kernel Density Estimation. RF random forest, RNN recurrent neural network, FCNN fully-connected neural network, SVM support vector machine, ARIMA autoregressive integrated moving average. **d** Nationwide prediction error distribution of each hour based on 2, 6 and 24-h ahead prediction. Each box includes 1917, 638, and 159 samples for solar energy and 4297, 1432, and 358 samples for wind energy. The lower/upper end of each box indicates the minimal/maximal value, and the lower and upper percentiles indicate 25% and 75%, respectively. The short blue line indicates the median, and the blue points show the outliers. There are blank areas for the 2-h and 6-h predictions since these two prediction tasks only contain 2 and 6 time periods, respectively.

the bubble size in Fig. 2d. The reason is that wind power prediction is conducted hour-by-hour, and the daily wind power generation is irregular and cannot reflect the hourly wind generation pattern. Regarding solar power, power generation varies periodically daily, and the characteristics of the hourly first-order difference could be masked by this daily periodicity.

Another significant factor influencing the prediction error is the peak ratio, which reflects the frequency of the tendency changes in the power output series, with CoD $R^2 = 0.967$ for the hourly wind output (Fig. 3a) and $R^2 = 0.558$ for the daily solar output (Fig. 3c). Similar to the first-order difference, wind and solar energy differ in their hourly and daily features. To further explore the impact of different power generation levels on the prediction error, we evenly divide the installed generation capacity into 10 intervals. We also select a representative province in each wind and solar energy category for detailed analysis. The representative wind energy provinces are TJ, SD, SX, and Gansu (GS); the representative solar energy provinces are BJ, JS, HB, and Inner Mongolia (IM). We express the peak distribution in each power generation interval as a frequency (Fig. 3b for wind energy and Fig. 3d for solar energy). Regarding wind energy, peaks in provinces with a large

prediction error, e.g., TJ: 13.6% and SD: 8.9% occur in both higher and lower power intervals, and the frequency fluctuates at 10%. However, in provinces with a small prediction error (SX: 5.4% and GS: 4.2%), peaks are concentrated in lower power intervals from 1 to 4, at 76.76% and 83.48%. In contrast, solar energy peaks are mainly located in higher power intervals, with the peaks in intervals above 4 accounting for 62.59%, 59.38%, 64.90%, and 89.61% in BJ, JS, HB, and IM, respectively.

## Temporal analysis of provincial prediction errors
We examine the PDF and prediction error in each province within the above 10 power generation intervals to analyze further the spatial characteristics of the prediction error (Fig. 4 and Supplementary Table 2). The results reveal that the more concentrated the PDF is within a certain interval, the smaller the prediction error within this interval. In terms of wind generation, the average prediction error within interval 1 in TJ is small (only 10.6%), and the PDFs within this interval are concentrated from intervals 1–4; in contrast, the prediction error within interval 8 reaches 21.5%, and the PDF within this interval is distributed across almost all intervals. The prediction error within each interval also reflects the variance and fluctuation magnitude within the

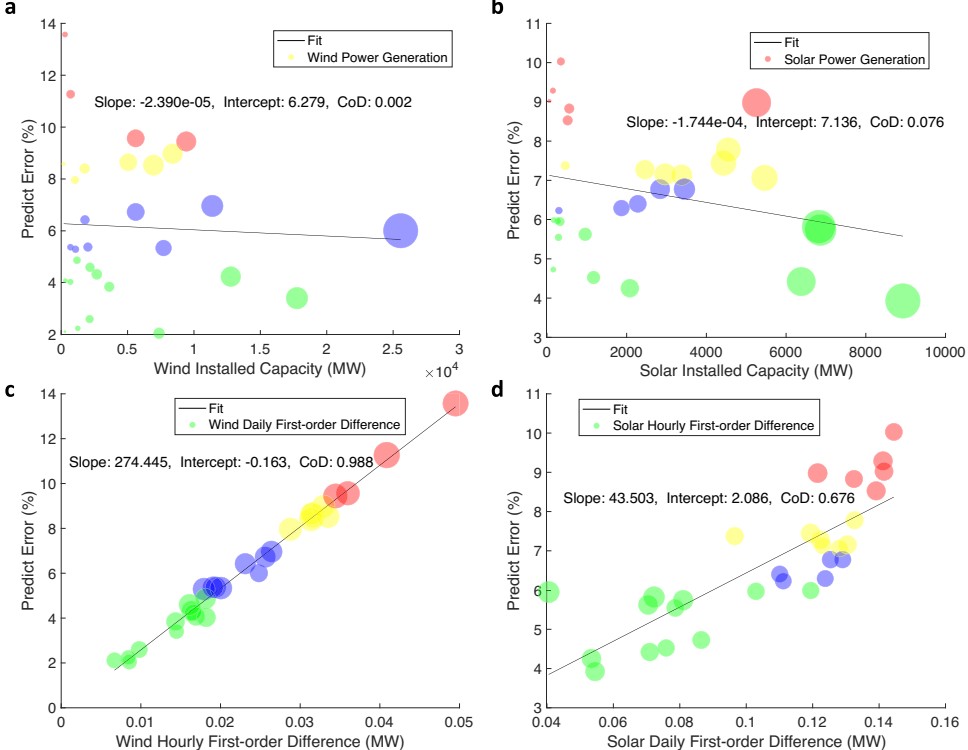

**Fig. 2 | Impacts of installed capacity, power generation and first-order difference of time series. a** wind installed capacity, (**b**) solar installed capacity, (**c**) wind hourly first-order difference, and (**d**), solar daily first-order difference. Here we use daily and hourly data to analyze solar and wind energy, respectively, which are presented in the x-axis. Each bubble indicates less influential factors, including wind or solar generation, wind daily first-order difference, and solar hourly first-order difference, respectively. The radius of each bubble is the value of each factor. The number of bubbles is 30, representing the 30 provinces of China, excluding Tibet (wind), Chongqing (solar), Hong Kong, Macao, and Taiwan. The black linear regression line fits the center of the bubbles, complemented by the slope, intercept, and coefficient of determination (CoD). The color of each bubble indicates the different categories: red−category with the largest prediction error; yellow−category with the second-largest prediction error; blue−category with the third-largest prediction error; green−category with the smallest prediction error. MW Megawatt.

interval. As shown in Fig. 4a, the average prediction error within interval 8 in TJ is larger than that within interval 1, and the fluctuation range within these two intervals is 0−72.1% with a variance of 404.2, and 0−32.9% with a variance of 134.5, respectively.

As illustrated in Fig. 4 and Supplementary Table 2, we also discover that most of the provinces with large prediction errors reach wind and solar prediction errors in high power intervals. The proportions of intervals above 5 in TJ for wind energy, SD for wind energy, SX for wind energy, BJ for solar energy, JS for solar energy, and HB for solar energy are 64.9%, 64.0%, 60.3%, 61.2%, 56.9%, and 53.4%, respectively. This phenomenon is more obvious for wind energy because solar power never occurs at full generation, and there is almost no solar power generation within intervals 9−10. Instead, the prediction errors in provinces with a small prediction error are distributed almost equally among all intervals, e.g., the wind prediction error within each interval in GS ranges from 8.3 to 22.8%. This occurs because high power generation generally exhibits peak or inflection points, which fluctuate wildly and are difficult to predict. The proportion of peaks within each interval is provided in Supplementary Table 3. Thus, the uncertainty of power generation can be intuitively assessed based on power generation.

We also analyze the seasonal characteristics of the generation uncertainty of solar and wind power on a provincial level. Here, we compare the provincial prediction error in spring, summer, autumn, and winter. Nationally, we determine that spring and summer are dominant seasons for wind uncertainty, accounting for 55.48% of the total prediction error (Fig. 5a), and spring and winter are dominant seasons for solar uncertainty, accounting for 57.6% of the total prediction error (Fig. 5c). The provincial characteristics are also similar, as illustrated in Fig. 5b, d. The wind uncertainties in TJ and SD in spring and summer account for 59.9% and 57.4%, respectively, of the total prediction error; the solar uncertainties in BJ, HB, and IM in spring and winter account for 60.4%, 58.0%, and 63.9%, respectively, of the total prediction error. This occurs because solar irradiation in summer and autumn is sufficient with fewer rainy days, resulting in more stable solar power generation and relatively accurate prediction results.

## Discussions

We provide an error-analysis benchmark for hourly wind and solar generation in 30 provinces of China with significance for research, industry, and policy decision-making. The proposed benchmark reveals statistical characteristics of wind and solar uncertainty, which is indispensable for academic research. First, it can help to build the PDF of wind and solar generation, providing scenario basis for stochastic economic dispatch[43]. Energy scheduling may also use renewable generation and consider their prediction errors as a probability distribution[44]. Second, the benchmark is applicable for robust optimization, because the best and worst-case operating conditions can be obtained through prediction results. It can also replace the assumed prediction errors to generate reasonable probability distribution and be used as expected forms in optimization formulations[45,46]. Third, risk assessment can also benefit from the benchmark, as the security region of power systems can be depicted based on the prediction results and errors[47]. Without our work, most of these research use assumed renewable generation and prediction error. In industry, the benchmark plays a critical role as a guiding reference for intuitive analysis of resource distributions and fluctuations, which could help to evaluate investment revenue and the risk of renewable projects. If

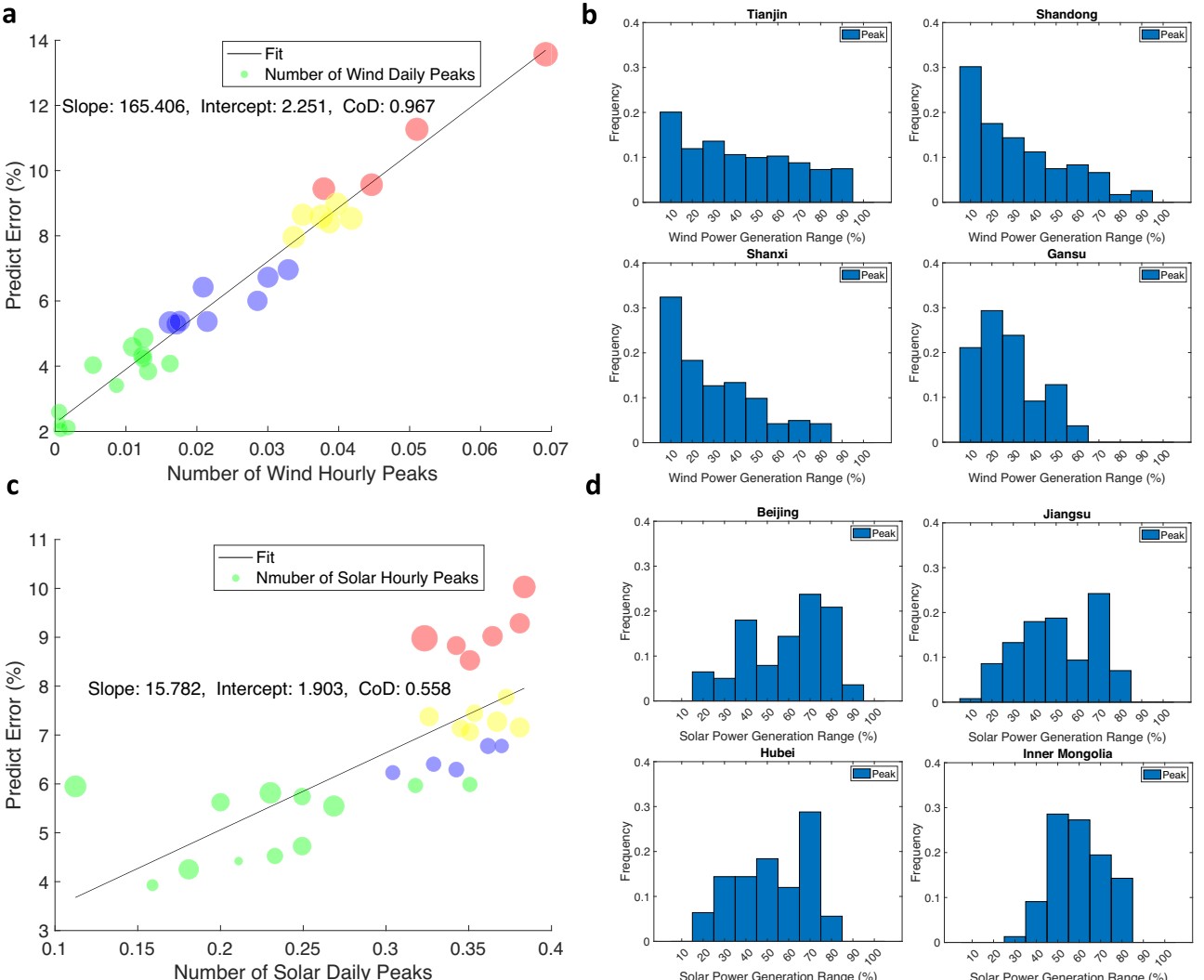

**Fig. 3 | Peaks distribution and the impact on the wind and solar power prediction errors. a** Influence of the wind hourly peaks. The radius of each bubble indicates the ratio of the wind daily peaks. **b** Wind hourly peak distribution in 10 power generation intervals for Tianjin (TJ), Shandong (SD), Shanxi (SX), and Gansu (GS). **c** Influence of the solar daily peaks. The radius of each bubble represents the ratio of the solar hourly peaks. **d** Solar daily peak distribution in 10 power generation intervals for Beijing (BJ), Jiangsu (JS), Hubei (HB), and Inner Mongolia (IM).

In (**a**) and (**c**), the number of bubbles is 30, representing the 30 provinces of China, excluding Tibet (wind), Chongqing (solar), Hong Kong, Macao, and Taiwan. The black linear regression line fits the center of the bubbles, complemented by the slope, intercept, and coefficient of determination (CoD). The color of each bubble indicates the different categories: red−category with the largest prediction error; yellow−category with the second-largest prediction error; blue−category with the third-largest prediction error; green−category with the smallest prediction error.

prediction errors are large and renewable generation is unstable, renewable projects will take more risks, and the investment should be reduced. In addition, policy-makers and system planners need information contained in the benchmark when determining development strategies for cleaner energy systems. An emergent and valuable issue entails the implementation of energy storage devices to mitigate the power balance stress in power systems with an increasing share of renewable resources[48,49], and the optimal sizing and setting processes of energy storage devices rely heavily on the spatial and temporal uncertainties of renewable generation. In this paper, we focus on the inherent uncertainty of renewable generation, and the forecasting errors are obtained merely by time-series analysis. In practice, the prediction errors of renewable generation may be impacted by more complicated factors such as weather forecasting quality and operational curtailment strategies. In some application scenarios, the forecasting tools may result in asymmetric errors conservatively. For instance, a system operator tends to forecast renewable generation conservatively for the sake of system reliability. These practical factors may lead to deviations in the distribution of the forecasting error, and

can be incorporated into the analysis by replacing the benchmark forecasting model with a more realistic one, which deserves an in-depth investigation in the future.

The statistical analysis indicates that the first-order difference and peak ratio of renewable generation are two primary influencing factors of prediction errors, both reflecting fluctuations in power generation. The wind prediction error is affected by the hourly power generation because the prediction model is employed based on the irregular hourly wind output. In contrast, the solar prediction error is affected by daily fluctuations since solar generation exhibits daily periodicity.

Our results reveal the provincial distribution of the uncertainty of wind and solar generation, indicating different priorities for renewable energy development in different areas. Some of the top 10 provinces with the largest wind prediction error are TJ, SH, JS, and AH, with values of 13.6%, 11.3%, 9.6%, and 8.4%, respectively. In contrast, the solar prediction error in these provinces is 9.0%, 10.0%, 7.1%, and 6.8%, respectively, which indicates that JS and AH should prioritize the development of solar energy due to the small prediction errors and fluctuations. SH and TJ are commercial provinces with small areas and

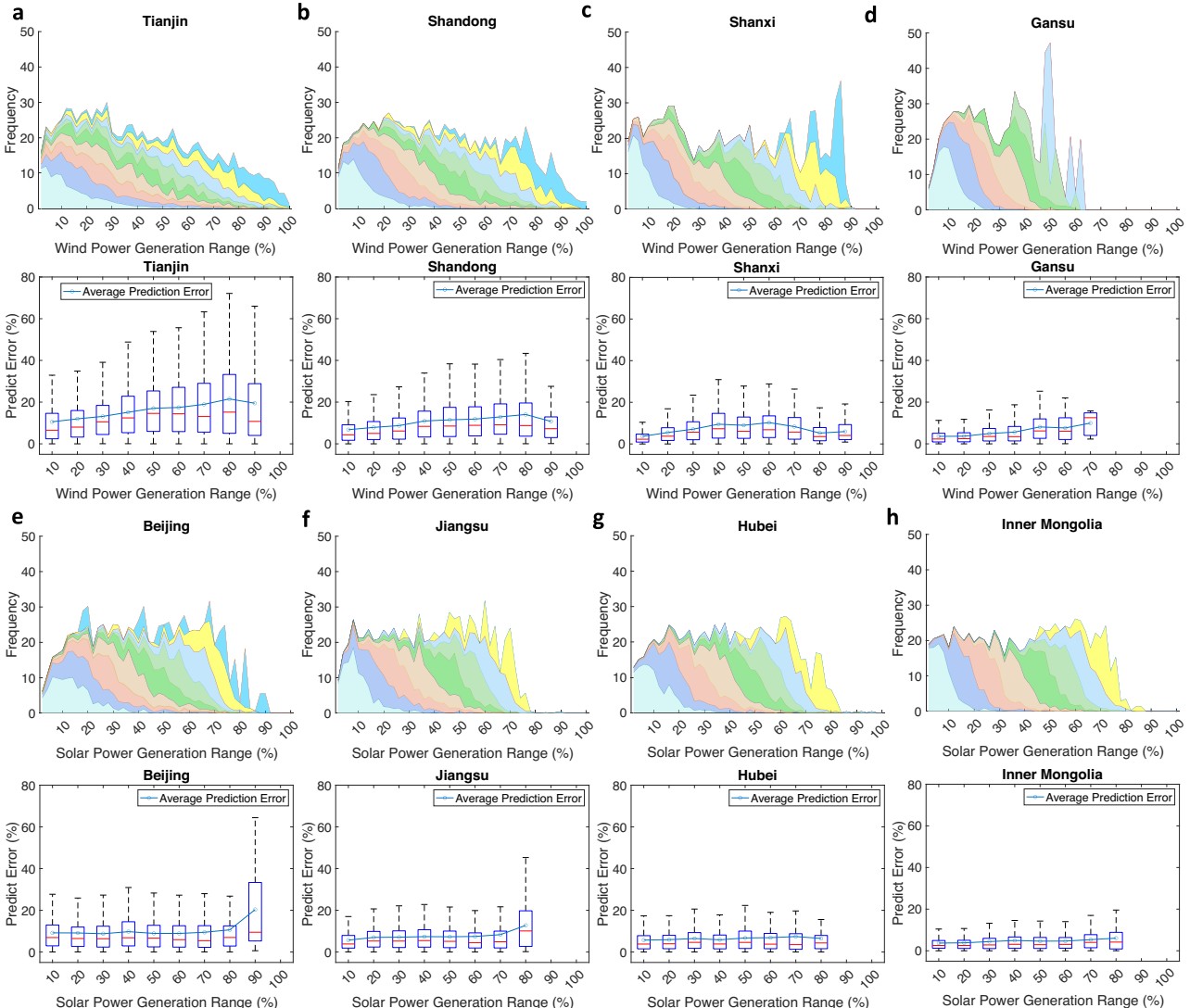

**Fig. 4 | Provincial probability distribution function (PDF) and prediction errors in each interval. a–d** The upper figures show the PDFs of wind prediction in Tianjin (TJ), Shandong (SD), Shanxi (SX), and Gansu (GS), and the lower figures show the wind prediction error in each interval. **e–h** The upper figure shows the PDFs of solar prediction in Beijing (BJ), Jiangsu (JS), Hubei (HB), and Inner Mongolia (IM), and the lower figure shows the solar prediction error in each interval. PDFs and box plots are missing in some intervals because the power generation does not reach that range of the installed capacity, such as TJ wind generation only covers 0–90% capacity. The PDFs plot indicates the distribution of predicted generation data. The x-axis indicates the predicted power generation range, and the color corresponds to the original power generation data in each generation range: pale turquoise: 0–10%; cornflower blue: 10–20%; dark salmon: 20–30%; burlywood: 30–40%; purple: 40–50%; pale green: 50–60%; light sky blue: 60–70%; yellow: 70–80%; deep sky blue: 80–90%. Different colors mean the frequency of a certain predicted power generation is composed of data from different power generation ranges. Each box shows the distribution of the prediction errors. The lower/upper end of each box indicates the minimal/maximal value, and the lower and upper percentiles indicate 25% and 75%, respectively. The short red line indicates the median and the bubble line indicates the average prediction error of each box.

are not suitable for wind and solar energy development. YN, Fujian, GS, Zhejiang (ZJ), and Guizhou (GZ) should develop wind energy due to their smallest prediction errors of 2.1%. 2.6%, 4.2%, 4.9%, and 3.8%, respectively. ZJ, SX, GZ, and SH are some of the top 10 provinces with larger solar prediction errors, namely, 7.1%, 7.2%, 7.4%, and 10.0%, respectively, while the wind prediction errors in ZJ, SX, and GZ reach 4.9%, 5.3%, and 3.8%, respectively, and the potential wind capacity factor for Sichuan and GZ is approximately 15–25%[10]. Therefore, wind energy development in these provinces is a recommended pathway to reduce the adverse impact of renewable generation on power system operation.

The temporal analysis demonstrates that renewable generation in spring exerts the greatest impact on the power system, requiring the proactive deployment of flexible resources. Combined with the spatial distribution, the solar prediction error in North China in winter

exhibits a large prediction error, ranging from 9.3 to 11.4%, with an average value of 10.4%, larger than the total prediction error of 3.9–10.0%, with an average value of 6.7%. As the Chinese government has issued the Electric Heating Policy to provide heat in North China in winter, the load demands in the power sector have increased significantly[50]. The flexibility-adjustable resources and volatility on the power source side exhibit inverse distributions, which have become a central problem in the consumption of renewable energy in these regions. In contrast, Southeast China achieves the smallest prediction error in regard to both wind and solar energy in winter, with average values of 2.8% and 5.1%, respectively. Additionally, existing research has suggested abundant offshore wind power resources in the area, with wind capacity factors higher than 50%, almost ranking at the top in China[10,11]. Due to the obvious seasonal distribution of offshore wind power, which dominates in spring and winter[51], wind power represents

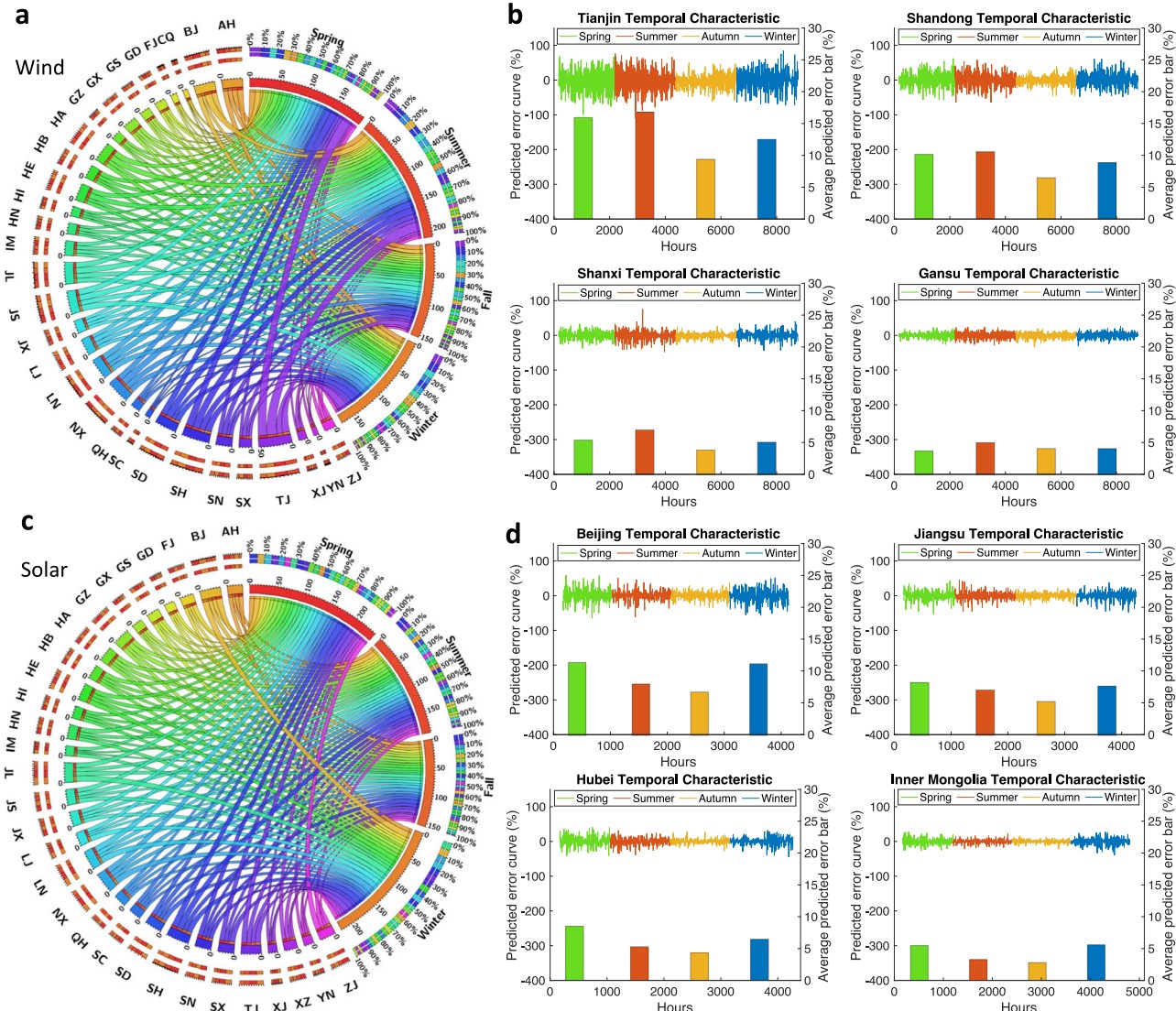

**Fig. 5 | Temporal analysis of wind and solar prediction errors. a** Wind, (**c**) solar prediction error in the 30 provinces in spring, summer, autumn, and winter. Each chord and arc represent the prediction error (%) between a province and the season, where the thickness is proportional to the level of prediction error. Regarding province arcs, each segment corresponds to the prediction error in each season; regarding season arcs, each segment corresponds to the prediction error in each province. The number next to the arc indicates the cumulative prediction error. Different colors in season arcs differentiate each province's influence. BJ Beijing, TJ Tianjin, HE Hebei, SX Shanxi, IM Inner Mongolia, LN Liaoning, JL Jilin, HL Heilongjiang, SH Shanghai, JS Jiangsu, ZJ Zhejiang, AH Anhui, FJ Fujian, JX Jiangxi, SD Shandong, HA Henan, HB Hubei, HN Hunan, GD Guangdong, GX Guangxi, HI Hainan, CQ Chongqing, XZ Tibet, SC Sichuan, GZ Guizhou, YN Yunnan, SN Shaanxi, GS Gansu, QH Qinghai, NX Ningxia XJ Xinjiang. **b** Hourly prediction error of wind power in TJ, SD, SX, and GS. **d** Hourly prediction error of solar in BJ, JS, HB, and IM. Curves indicate hourly prediction errors (left axis), and bars indicate average prediction errors (right axis) in the four seasons: Green−spring; red−summer; yellow−autumn; and blue−winter.

a suitable alternative resource to offset the winter load peak in North and Northeast China.

Based on the prediction error analysis, we summarize two policy suggestions for China. First, the government should provide adequate policy support and incentives to encourage wind energy development in the Southwestern and Central areas of China and solar energy development in the areas of Southwest and Northwest China. These areas experience limited fluctuations in wind and solar generation, around 2.1–6.4% and 4.3–7.4%, reducing the adverse impact on the power system. However, the current installed capacities in these regions are insufficient, even lower than East area with less land. Second, the government should plan interprovincial energy transmission in the space dimension to reduce the winter load peak in North China and reduce the adverse impact of renewable energy. As concluded, the wind and solar fluctuations in North China are notable, accounting for

28.1% and 25.0%, respectively, of the total prediction error in China, especially during winter, with a proportion of 27.4% and 27.7%. However, during spring and summer, much energy consumption can be satisfied by renewable energy, resulting in an unbalance in different seasons and requiring additional energy sources. As such, the government should improve the power system infrastructure, systematically evaluate potential transmission projects, and plan additional power lines according to the resource and load distribution.

## Methods

### Wind and solar output data
Hourly wind and solar output data for 2016 pertaining to 30 provinces of China are retrieved from previous work[11], except for Tibet wind, Chongqing solar, Taiwan, Hong Kong, and Macao. The dataset contains 8760 h of wind and solar output data, and wind and solar installed

capacity data for these 30 provinces are included. We denote the hourly wind output as $W_{i,t+1,0}$ and the hourly solar output as $S_{i,t+1,0}$, where $i$ and $t$ are province and time slot indices, respectively, for $i \in [1,N], t \in [1,T]$, $N = 30$, and $T = 8760$. As previously mentioned, daily wind and solar output data are also required for the analysis, which can be calculated as Eqs. (1)-(2):

$$W_{\text{Day},i,c,0} = \max(W_{i,t,0}, W_{i,t+1,0}, \cdots W_{i,t+23,0}), t = 24 \cdot (c-1) \quad (1)$$

$$S_{\text{Day},i,c,0} = \max(S_{i,t,0}, S_{i,t+1,0}, \cdots S_{i,t+23,0}), t = 24 \cdot (c-1) \quad (2)$$

where $S_{\text{Day},i,c,0}$ and $W_{\text{Day},i,c,0}$ are the daily solar and wind output, respectively, of province $i$ in time slot $t$, and $c$ is a day index, for $c \in [1,C]$ and $C = 365$.

## Benchmark prediction model

Time series prediction is based on historical data, among which the autoregressive (AR), moving average (MA), and autoregressive moving average (ARMA) techniques are typical methods to study stationary time series and are suitable for a large number of problems. However, the fluctuations in wind and solar energy indicate that their power generation involves a nonstationary time series with a time-varying mean value and variance, which is difficult to study with these methods. Thus, to predict nonstationary sequences, the ARIMA prediction model is introduced by Box-Jerkins. Considering a certain number of differences in the ARIMA prediction model, wind and solar power generation series can be converted into a stationary series, convenient for prediction analysis. In the literature, the ARIMA model is widely used in short-term renewable forecasting and is validated to yield satisfactory results.

In prediction model construction, it is necessary to first determine whether the series is stationary. If the series is not stationary, it should be differentiated until the series meets the stationarity requirements. Suppose the real wind and solar power generation series are $Y_t$, the differential order can be denoted by $d$, and the differential process can be expressed as Eq. (3):

$$X_t = (1 - B)^d Y_t, \text{ADFtest}(X_t) = 1, \quad (3)$$

where $X_t$ is the stationary series of the original real data, $B$ is the lag operator, and ADFtest $= 1$ passes the stationarity test. Except for the differential order $d$, the ARIMA model should also determine the autoregressive order $p$ and moving average order $q$, and the ARMA model for $X_t$ can be expressed as Eq. (4):

$$\left(1 - \sum_{i=1}^{p} \varphi_i B^i\right) X_t = \mu_0 + (1 - \sum_{i=1}^{q} \mu_i B^i) \alpha_t, \quad (4)$$

where $\varphi_i$ and $\mu_i$ are the autoregressive parameter and moving average parameter, respectively, $\alpha_t$ is white noise with a mean of 0, $\mu_0$ is a deterministic trend quantity greater than 0, and $B^i$ is the $i$th power of $B$. Via the use of the prediction model, we can obtain the predicted series $X_{\text{predict},t}$, which is a differential series of the predicted wind and solar power generation. Thus, the predicted power generation can be obtained through Eq. (5):

$$Y_{\text{predict},t} = (1 - B)^{-d} X_{\text{predict},t}, \quad (5)$$

where $Y_{\text{predict},t}$ denotes the predicted results of the ARIMA-based prediction model, and in this paper, this variable indicates the wind and solar output.

There are three major parameters of the ARIMA-based prediction model: differential order $d$, autoregressive order $p$, and moving average order $q$. Parameter $d$ is determined based on the minimum number of differences required to obtain a stationary time series. The $d$ value is

generally smaller than three because the greater the difference order, the more information would be lost[52]. It should be noted that parameter $d$ is completely determined by the properties of the original sequence, while the selection of $p$ and $q$ should consider the overall prediction effect. In general, $p$ and $q$ should remain within 1/5 of the length of the input data. Due to the large amount of wind and solar power generation data in each province in one year, usually 8760 h, we separate multiple prediction windows for each province and used the moving window method to predict wind and solar power generation. At present, the methods for $p$ and $q$ determination usually include the Akaike information criterion (AIC) and Bayesian information criterion (BIC), but the optimal parameter configuration can only be provided for a single prediction window. To unify the prediction models with the different prediction windows in the same provinces and minimize the prediction error, we randomly select 5 weeks of data throughout the year as a sample and traverse $p$ and $q$ for each province to obtain the best parameters with the minimum prediction error. The detailed parameters for each province are listed in Supplementary Table 4.

Other parameters, such as the autoregressive parameter $\varphi_i$ and moving average parameter $\mu_i$, can vary with the input data. These two parameters are determined by the autocorrelation coefficient and autocovariance, respectively, which can be obtained with the Yule–Walker estimation, least squares estimation or maximum likelihood estimation method[53]. In this paper, we build the ARIMA-based prediction model, and all the parameters except $p$, $d$, and $q$ could be automatically generated.

In this paper, we set 6 h as the prediction time scale and 168 h as the input data dimension to predict wind and solar power generation. The reason is that 6 h-ahead forecast of renewable generation is widely used for power system scheduling and electricity trading in practice. The 6 h-ahead forecast also results in moderate errors that can serve as a benchmark for the uncertainty analysis.

## Comparative prediction models

In this paper, we compare four prediction methods including RF, FCNN, RNN, and SVM. These four methods are all sample-based prediction approaches. We begin by constructing the samples using 168-h wind and solar generation data as input features and extracting subsequences of 2, 6, and 24 h as output for 2-h, 6-h, and 24-h step predictions, respectively. The RF method employs a tree-based prediction model that builds multiple decision trees during training. The structure of the decision trees is determined by parameters such as tree depth, the number of trees, and the maximum number of features considered when splitting nodes. The FCNN method utilizes a network structure consisting of interconnected perceptron. Each time slot's generation data serves as an input feature for the FCNN, and the predicted generation is the output. The network structure is designed based on factors such as regularization, batch size during training, learning rate, and the number of neurons in each layer. The RNN is a neural network structure specifically designed for time series data, incorporating hidden variables to carry information from previous time slots. Similar to the FCNN, the RNN's network structure is determined by parameters including the number of neurons, batch size, and learning rate. The SVM is an initial machine learning method employed to separate the dataset. The SVM solves an optimization problem to find an optimal hyperplane. Key considerations for SVM include regularization parameters, the margin of tolerance around predicted regression values, and the influence attributed to each sample. Further details on the network parameters and the tuning process can be found in the Supplementary Note and Supplementary Table 5.

## Prediction error calculation

In this paper, the prediction error of wind and solar energy could be calculated as the unit megawatt (MW) prediction error. When using the ARIMA-based benchmark prediction model, we could obtain the

predicted wind and solar energy generation, and the prediction error can then be calculated as Eq. (6):

$$\varepsilon_{\text{W},i,t} = \frac{W_{i,t,*} - W_{i,t,0}}{C_{\text{W},i}} \cdot 100\%, \varepsilon_{\text{S},i,t} = \frac{S_{i,t,*} - S_{i,t,0}}{C_{\text{S},i}} \cdot 100\%, \quad (6)$$

where $\varepsilon_{\text{W},i,t}$ and $\varepsilon_{\text{S},i,t}$ are the wind and solar prediction error in province $i$ in time slot $t$, $W_{i,t,*}$ and $S_{i,t,*}$ are the predicted wind and solar output, respectively, of province $i$ in time slot $t$, and $C_{\text{W},i}$ and $C_{\text{S},i}$ are the wind and solar installed capacities, respectively, in province $i$. When determining the prediction error in a given province, we calculate the average value over 8760 h.

### First-order difference
The first-order difference can be used to assess the variation in discrete time-series data. With the use of the first-order difference, we can obtain the increment in the original data, which can reflect gradient information. In this paper, prediction is conducted hour-by-hour, and the prediction accuracy is primarily determined by the hourly change in the generation data. Thus, in terms of wind energy, we use the first-order difference of hourly wind generation data to measure the hourly change, which can be calculated as Eq. (7):

$$F_{\text{H},i,t} = \frac{W_{i,t+1,0} - W_{i,t,0}}{C_{\text{W},i}}, \quad (7)$$

where $F_{\text{H},i,t}$ is the hourly first-order difference in province $i$ in time slot $t$ and $W_{i,t+1,0}$ and $W_{i,t,0}$ are the real wind energy generation in time slots $t+1$ and $t$, respectively. When evaluating the hourly first-order difference in a province, we calculate the average value over 8760 h.

Regarding solar energy, power generation exhibits daily periodicity, so we use daily solar energy generation data to measure the fluctuation, which can be expressed as Eq. (8):

$$F_{\text{Day},i,c} = \frac{S_{\text{Day},i,c+1,0} - S_{\text{Day},i,c,0}}{C_{\text{S},i}}, \quad (8)$$

where $F_{\text{Day},i,c}$ is the daily first-order difference in province $i$ on day $c$. We also calculate the average value over 365 days to evaluate the solar energy fluctuations in a given province.

### Analysis and calculation of the peak ratio
In this paper, we use the peak ratio to evaluate the prediction error. It should be noted that all the prediction methods learn the variation tendency of a given data series to predict future data. The easier a tendency is to learn, the more accurate the prediction. Thus, we aim to obtain a feature that could indicate the change in tendency to better measure the prediction error. The peaks of series data indicate inflection points, with previous data exhibiting an upward tendency and subsequent data exhibiting a downward tendency, which is a key feature reflecting the tendency change.

In regard to wind energy, we use four consecutive time slots to determine hourly peaks and traverse the time series to find all peaks, i.e., $t = t + 1$. The power generation in these four time slots should satisfy the following conditions to reach a peak: the first three hours should continuously increase, the first three hours should increase by more than 10% of the installed capacity, and the fourth hour should decrease, which can be expressed as Eqs. (9)–(11):

$$P_{\text{H},i,t} = 1, W_{i,t,0} - W_{i,t-1,0} < 0, W_{i,t-1,0} - W_{i,t-2,0} \geq 0, W_{i,t-2,0}$$
$$- W_{i,t-3,0} \geq 0, W_{i,t-1,0} - W_{i,t-3,0} \geq 0.1 \cdot C_{\text{W},i}, \quad (9)$$

$$P_{\text{N,H},i} = \sum_{t \in T} P_{\text{H},i,t}, \quad (10)$$

$$P_{\text{R,H},i} = P_{\text{N,H},i} / T \quad (11)$$

where $P_{\text{H},i,t}$ denotes the hourly peaks in province $i$ in time slot $t$, $P_{\text{N,H},i}$ is the number of hourly peaks in province $i$, and $P_{\text{R,H},i}$ is the ratio of hourly peaks in province $i$. We also calculate the average value over 8760 h to evaluate the wind energy fluctuations in each province.

Regarding solar energy, we use daily power generation data to obtain daily peaks. Similar to the hourly peak calculation, four consecutive days are chosen to determine peaks, and similar conditions should be satisfied, which can be expressed as Eqs. (12)–(14):

$$P_{\text{Day},i,c} = 1, S_{\text{Day},i,c,0} - S_{\text{Day},i,c-1,0} < 0, S_{\text{Day},i,c-1,0} - S_{\text{Day},i,c-2,0} \geq 0, S_{\text{Day},i,c-2,0}$$
$$- S_{\text{Day},i,c-3,0} \geq 0, S_{\text{Day},i,c-1,0} - S_{\text{Day},i,c-3,0} \geq 0.1 \cdot C_{\text{S},i}, \quad (12)$$

$$P_{\text{N,Day},i} = \sum_{c \in C} P_{\text{Day},i,c}, \quad (13)$$

$$P_{\text{R,Day},i} = P_{\text{N,Day},i} / C, \quad (14)$$

where $P_{\text{Day},i,c}$ is the daily peak in province $i$ on day $c$, $P_{\text{N,Day},i}$ is the number of daily peaks in province $i$, and $P_{\text{R,Day},i}$ is the ratio of daily peaks in province $i$. The average value over 365 days is also calculated to express the solar energy fluctuations in each province.

## Data availability
The source data underlying Figs. 1–5 and Supplementary Figs. 1-4, including the data of provincial wind and solar power generation of the 30 provinces in China, are provided as a Source Data file. Other data used in this study are available from the authors upon reasonable request. Source data are provided with this paper.

## Code availability
The code used in this study is available from the authors upon reasonable request.

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

## Acknowledgements

We thank the National Key Research and Development Program of China no. 2022YFB2405600 for supporting J.W. and G.H. and the National Natural Science Foundation of China under grant No. 72241429, No. 72271008, No. 72243007, and No. 52277092 for supporting J.S., G.H., X.L., and J.W. We also acknowledge the support of State Grid Corporation of China, State Grid Jiangsu Electric Power Co., LTD. and State Grid Wuxi Power Supply Company.

## Author contributions

J.W., X.L., N.L., J.M., J.S. and G.H. conceived and designed the research. J.W., L.C., C.W.T., C. L. and G.H. developed the framework and formulated the theoretical model. J.W., L.C., X.L. and E.D. carried out the data search. L.C., Z.T. and M.S. carried out the simulations. J.W., X.L., G.H. and C.W.T. conducted the prediction-error analysis. J.W., L.C., Z.T., E.D., N.L., J.M., M.S., C.L., J.S., X.L., C.W.T. and G.H. contributed to the discussions on the method and the writing of this article.

## Competing interests

The authors declare no competing interests.
