## [Peer Review file · Nature Communications]

Inherent Spatiotemporal Uncertainty of Renewable Power in ChinaREVIEWER COMMENTS

Reviewer #1 (Remarks to the Author):

Deploying renewable energy is an essential component of efforts to reduce greenhouse gas emissions. The variability of wind and solar energy challenges the decarbonization of power system. This paper attempts to quantify the wind and solar energy uncertainty by analyzing their prediction errors. Such works are becoming increasingly important, and it can be beneficial to the power system operators or capacity planners. However, the manuscript in its current form demonstrates a lot of room for further improvement. Major revision is required.

My overall impression of the paper is that the author put a lot of work and effort into this work and also in writing this manuscript which I definitely would like to acknowledge here. However, this paper still needs substantial improvements. My comments below outline possible strategies for revision, although the authors may choose to address these concerns in other ways.

1. To me, the major flaw of this study is the method to calculate prediction error. The variables used to calculate the prediction error of wind and solar energy include the predicted and actual output. First, maybe I do not understand the authors, but it seems the authors do not clearly provide the way to obtain the actual output. In lines 427-428, the authors say that actual output data for 2018 are retrieved from work 11, but work 11 was published in 2016. Also, work 11 was only about wind power. Work 11 provides the calculated wind output, which is the result of wind resource and a typical GE turbine; this work does not provide actual renewable output. Anyway, I want to know how the authors obtain actual output, and whether it is truly actual output (or is it integrated renewable energy?). Second, regarding the predicted/forecasted output, there are many methods to obtain this variable. The authors use an ARIMA-based prediction model to generate predicted output, which may be different to China's practice to predict/forecast renewable output. To my understanding, the predicted/forecasted output is a result of many different methods, including estimation from wind resources like the way in Work 11. System operators or planners generally use predicted/forecasted output from resources' estimation. I suggest the authors at least compare the output differences from different methods, and provide a reason to use ARIMA-based prediction model. Also, there are many different time-scales of prediction in practice, for example, day-ahead, 6-hours ahead, 2-hours ahead, 15-mins ahead. Different time scales have different forecast errors, and the predict/forecast accuracy changes accordingly. The authors should have a thorough discussion about this. By the way, there are many newly published papers which estimate renewable energy potential and variation. I suggest the authors conduct a more thorough literature review, which would be beneficial to highlight the value of this paper.

2. In this paper, the method to calculate the carbon emissions effects of prediction error is coarse, and I suggest the authors delete this part; the authors may have different ways to improve this part. The

predicted/forecasted output can be higher or lower than the actual output, and different situations bring different impacts on carbon emissions. Without a careful power system dispatch analysis, the carbon emissions effects are meaningless. Although the authors said in Supplementary Figure 1 they provide an aggressive estimation, the results may not be an upper bound if we consider startup and no-load emissions and power system operations.

3. To my understanding, in practice system operators and planners tend to predict/forecast output conservatively. In this way, if the actual output is higher than the predicted one, the energy can be curtailed. This is due to the consideration of power system reliability. There are also many other factors affecting predict/forecast output, such as renewable resources' forecast accuracy. I suggest the authors at least provide some discussions on this, or even analyze its impact.

4. Fig 1d provides carbon emissions and the proportion caused by prediction errors. Which is the source of China's power sector's CO₂ emissions and installed capacity of different generation technologies from 2021 to 2030 in Fig 1d? I think there are different trajectories and I suggest the authors provide a sensitive analysis in Fig 1d, or change this figure to other forms. By the way, in line 153 it is CO₂ emissions, not C02 emissions.

5. Fig 2, is it wind daily (in Fig. 2c) or hourly (in caption) difference? I think it should be hourly. Is it solar daily (in caption) or hourly (in Fig. 2d) difference? It should be daily.

Reviewer #2 (Remarks to the Author):

This is an impressive study on an important topic, which certainly merits publication. The authors examined the statistical characteristics of the inherent uncertainties of renewable generation arising from natural randomness. From a very fundamental and novel perspective, the numerical results reveal a pattern of renewable's spatiotemporal distributions across 30 provincial regions in China. Additionally, the authors try to answer a public interest as to how to quantify the extra carbon emissions induced by renewable uncertainty. Such a paradigm should work for similar knowledge discovery in other renewable-penetrated countries as well. The proposed method and database could potentially provide important references for stochastic-aware research and applications, which deserves a series of follow-up studies from different fields. In this instance, I suggest minor revisions as follows:

1. I was left a little unclear about the estimation for renewable uncertainty-induced carbon emissions. Can the authors attempt a wider context to explain the physical meaning of the estimation method? Also, how do the authors consider the uncertainty in the future?

2. It's interesting that the prediction error is highly dependent on the first-order difference and peaks of renewables. However, why do the authors define different calculation methods for wind and solar separately? In addition, the regression functions should be presented in Fig. 2 and 3.

3. Please further explain why the probability distribution functions of some intervals are absent in Fig. 4. For example, Beijing has ten intervals but Inner Mongolia only has seven. Do the authors have any special consideration?

4. Some other minor comments should include: i) The figures have to be further polished to comply with publication requirements. For example, the legends and curves in Fig. 5 have overlaps. Also, it is ambiguous to justify the relationship between y-axis and the curve/bar. ii) Please further improve the quality of the policy implication part to demonstrate the role of the main contribution.

Reply to the Comments

Reviewer Comments:

Reviewer #1 (Remarks to the Author):

Deploying renewable energy is an essential component of efforts to reduce greenhouse gas emissions. The variability of wind and solar energy challenges the decarbonization of power system. This paper attempts to quantify the wind and solar energy uncertainty by analyzing their prediction errors. Such works are becoming increasingly important, and it can be beneficial to the power system operators or capacity planners. However, the manuscript in its current form demonstrates a lot of room for further improvement. Major revision is required. My overall impression of the paper is that the author put a lot of work and effort into this work and also in writing this manuscript which I definitely would like to acknowledge here. However, this paper still needs substantial improvements. My comments below outline possible strategies for revision, although the authors may choose to address these concerns in other ways.

Comment 1.1

To me, the major flaw of this study is the method to calculate prediction error. The variables used to calculate the prediction error of wind and solar energy include the predicted and actual output. First, maybe I do not understand the authors, but it seems the authors do not clearly provide the way to obtain the actual output. In lines 427-428, the authors say that actual output data for 2018 are retrieved from work 11, but work 11 was published in 2016. Also, work 11 was only about wind power. Work 11 provides the calculated wind output, which is the result of wind resource and a typical GE turbine; this work does not provide actual renewable output. Anyway, I want to know how the authors obtain actual output, and whether it is truly actual output (or is it integrated renewable energy?).

Response:

We appreciate the valuable comments and suggestions from this reviewer.

We apologize for the mistake of writing 2016 as 2018 in the original manuscript. We have carefully checked the data source. The wind and solar generation data used in this paper are actually from 2016.

Admittedly, the data are not collected from the measured generation output of wind and solar power. The data quality of actual renewable output obtained from power grid company in China is very poor. Bad data, abnormal data and data loss are often observed. Consequently, instead of using the poor actual renewable output, real meteorological data is generally used to generate renewable output. In this paper, renewable output data are generated using the method developed in [11]. This method obtains real meteorological data over the years [R1], and generates the power output data of renewable energy through the typical numerical weather of the provincial capital city. Based on the simulated renewable power data, we test the prediction errors of different forecast methods to examine the uncertainty of renewable generation, which reveals the inherent characteristics of realistic meteorological data.

[R1] Rienecker, M. M. et al. The GEOS-5 Data Assimilation System-Documentation Versions 5.0.1, 5.1.0, and 5.2.0, 118 (NASA, 2007).

Action:

In the revised manuscript, the data source has been revised from 2016, and detailed modifications are shown below:

“Hourly wind and solar output data for 2016 pertaining to 30 provinces of China are retrieved from previous work, except for Tibet wind, Chongqing solar, Taiwan, Hong Kong, and Macao.”

Second, regarding the predicted/forecasted output, there are many methods to obtain this variable. The authors use an ARIMA-based prediction model to generate predicted output, which may be different to China’s practice to predict/forecast renewable output. To my understanding, the predicted/forecasted output is a result of many different methods, including estimation from wind resources like the way in Work 11. System operators or planners generally use predicted/forecasted output from resources’ estimation. I suggest the authors at least compare the output differences from different methods, and provide a reason to use ARIMA-based prediction model.

Response:

We appreciate the valuable comments and suggestions from this reviewer.

We agree with this reviewer that there is broad spectrum of methods for renewable power forecasting and the forecasting task may focus on diversified time scales such as day-ahead, 6-hour-ahead, 2-hour-ahead, 15-mins ahead, etc. The target of the present work is not on developing new forecasting methods. Instead, we are motivated to examine the inherent uncertainty of renewable power generation. ARIMA is one of the most fundamental models for time-series forecasting and is also found to be effective for short-term wind and solar generation forecasting. Hence, we take the ARIMA as a rule-of-thumb method to analyze the uncertainty of renewable generation time series.

With the valuable comments from this reviewer, we recognize that the forecasting errors from different methods on diversified time scales should be compared to validate the conclusion of this work. In the revised manuscript, we compare the performance of the ARIMA-based forecasting method with another 4 methods, i.e., (1) random forest (RF), (2) recurrent neural network (RNN), (3) fully-connected neural network (FCNN) and (4) support vector machine

(SVM). Hyper-parameters of these methods are finely tuned to improve their forecasting performance. Detailed parameters of different forecasting methods are presented in Supplementary Table 4 of the Supplementary Information.

The prediction errors of the different methods are shown in Fig. r1. The results show that different methods result in different prediction results. The RF, FCNN, and ARIMA outperforms the FCNN and SVM in terms of the forecasting accuracy. For wind energy prediction, the ARIMA not only yields the lowest prediction error, but also has a more concentrated error distribution compared to other methods. For solar energy prediction, the ARIMA performs comparably well as the RF and RNN. Further details regarding the nationwide prediction errors are provided in Table r1. In terms of nationwide prediction error, all methods yield similar results, ranging from 6% to 9%, suggesting that the choice of method has limited influence on the nationwide forecasting accuracy. To this end, the ARIMA is used as a benchmark prediction model in this paper.

Fig. r1 Prediction error distribution across 30 provinces obtained by different methods.

The smoothed curve in the left and right parts represents the prediction error density function across 30 provinces of wind and solar energy, respectively. The short black line in the middle of each shape is the median value of the data distribution, which visualizes the central tendency of the data distribution of each method. The algorithm used to fit the density function is Kernel Density Estimation.

Table r1 Mean value in Fig. r1

Method	Solar Prediction error (%)	Wind Prediction error (%)
ARIMA	6.16	6.70
FCNN	6.79	9.53
RF	5.44	7.60
RNN	5.18	7.16
SVM	7.18	9.20

Action:

The original Fig. 1c has been replaced by Fig. r1, and the manuscript has been revised as follows:

“We compare the prediction errors of various methods, including random forest (RF), recurrent neural network (RNN), fully-connected neural network (FCNN), and support vector machine (SVM), for predicting nationwide renewable energy output. The results are presented

in Fig. 1c. Our observations indicate that although each method demonstrates varying prediction error distributions across different provinces, the overall nationwide prediction errors remain similar among all methods, ranging from 6% to 9%. Further details can be found in the SI. Notably, ARIMA and RNN exhibit similar prediction errors and outperform other methods, benefiting from their inherent ability to effectively handle time series data. In the following part of this paper, we focus on the prediction error with the ARIMA model as a benchmark method.”

Also, there are many different time-scales of prediction in practice, for example, day-ahead, 6-hours ahead, 2-hours ahead, 15-mins ahead. Different time scales have different forecast errors, and the predict/forecast accuracy changes accordingly. The authors should have a thorough discussion about this.

Response:

We appreciate the valuable comments and suggestions from this reviewer.

In the revised manuscript, we also analyze the influence of forecasting time scale on the accuracy. Fig. r2 shows the prediction error distribution by time horizon for the 30 provinces in China. We observe that prediction error increases with the prediction time scale, with a 2-hour prediction resulting in a 3.40% error for solar and a 2.83% error for wind, a 6-hour prediction resulting in a 6.14% error for solar and a 6% error for wind, and a 24-hour prediction resulting in a 9.25% error for solar and a 10.86% error for wind. The prediction error increases with the time slots in the forecasting time scale, especially for the ending hours.

Fig. r2 Nationwide prediction error distribution of each hour based on 2, 6 and 24-hour ahead prediction. Each box includes 1917, 638, and 159 samples for solar energy and 4297, 1432, and 358 samples for wind energy. The lower/upper end of each box indicates the minimal/maximal value, and the lower and upper percentiles indicate 25% and 75%, respectively. The short blue line indicates the median, and the blue points show the outliers. There are blank areas for the 2-hour and 6-hour predictions since these two prediction tasks only contain 2 and 6 time periods, respectively.

It should be noted that the 6-hour-ahead forecasting is widely used in practical power system scheduling and electricity trading since the majority of the numerical weather prediction

(NWP) tools are updated 4 times each day (i.e., a 6-hour-ahead time frame), such the Local Prediction Model for Asia, Asian Dust Aerosol Model 3 (ADAM3), and the NOAA Global Ensemble Forecast System (GEFS). Additionally, the 6-hour-ahead forecast also results in moderate errors that can serve as a benchmark for the uncertainty analysis. A 2 hours-ahead prediction time scale produces better results, but it is more suitable for real-time dispatch. Day-ahead prediction requires the prediction method to capture relatively long-term generation characteristics. To this end, the 6-hour-ahead forecasting is used as the benchmark for detailed spatial-temporal analysis in this paper.

Action:

The original Fig. 1d has been replaced by Fig. r2, and the manuscript has been revised as follows:

“Moreover, we examine the impact of the prediction time scale on the distribution of nationwide prediction errors for both wind and solar energy, as illustrated in Fig. 1d. We observe that prediction error increases with the prediction time scale, with a 2-hour prediction resulting in a 3.40% error for solar and a 2.83% error for wind, a 6-hour prediction resulting in a 6.14% error for solar and a 6% error for wind, and a 24-hour prediction resulting in a 9.25% error for solar and a 10.86% error for wind. A detailed analysis of each hour's prediction error reveals that the error mainly originates from the ending periods, e.g., during 5-6 hours for the 6-hour ahead predictions and during 15-24 hours for the 24-hour ahead predictions.”

By the way, there are many newly published papers which estimate renewable energy potential and variation. I suggest the authors conduct a more thorough literature review, which would be beneficial to highlight the value of this paper.

Response:

We appreciate the valuable comments and suggestions from this reviewer.

We agree with the reviewer that the estimation of renewable generation potential and variation has attracted lots of attention in the existing literature. However, there is still a lack of spatial-temporal analysis of the uncertainty characteristics of renewable generation, which is important for the planning, investment, operation, and policy making for the integration of volatile renewable energy. In the revised manuscript, we improve the literature review by analyzing the recent publications on the estimation of renewable energy potential and variation. **Action:**

To highlight the motivation and contribution of this work, we improve the literature review by adding the following contents and references.

“However, renewable energy resources rely on weather conditions and thus are highly unstable, posing great challenges to accurate and reliable prediction. Some studies have examined the uncertainty of solar and wind power equipped with energy storage to assess their potential to meet future electricity demand²⁰. Prediction methods such as linear regression models and XGBoost have been utilized to forecast the uncertainty of wind and solar generation in specific regional areas, considering seasonal or yearly analyses²¹⁻²². However, limited research has focused on analyzing the spatiotemporal uncertainty distributions of renewable energy²³⁻²⁴. There are research gaps in terms of error analysis benchmarks that consider longterm, high-granularity, and nationwide scales of wind and solar output prediction, particularly

within the context of China.

20. Tong, D., Farnham, D.J., Duan, L. et al. Geophysical constraints on the reliability of solar and wind power worldwide. *Nat Commun.* 12, 6146 (2021).

21. Zeng, P., Sun, X. & Farnham, D.J. Skillful statistical models to predict seasonal wind speed and solar radiation in a Yangtze River estuary case study. *Sci Rep.* 10, 8597 (2020).

22. Joshi, S., Mittal, S., Holloway, P. et al. High resolution global spatiotemporal assessment of rooftop solar photovoltaics potential for renewable electricity generation. *Nat Commun.* 12, 5738 (2021).

23. Yin J, Molini A, Porporato A. Impacts of solar intermittency on future photovoltaic reliability. *Nat Commun.* 11, 1-9 (2020).

24. Anadón D. L., Baker E. & Bosetti V. Integrating uncertainty into public energy research and development decisions. *Nat Energy.* 2, 17071 (2017).”

Comment 1.2

In this paper, the method to calculate the carbon emissions effects of prediction error is coarse, and I suggest the authors delete this part; the authors may have different ways to improve this part. The predicted/forecasted output can be higher or lower than the actual output, and different situations bring different impacts on carbon emissions. Without a careful power system dispatch analysis, the carbon emissions effects are meaningless. Although the authors said in Supplementary Figure 1 they provide an aggressive estimation, the results may not be an upper bound if we consider startup and no-load emissions and power system operations.

Response:

We appreciate the valuable comments and suggestions from this reviewer.

As the reviewer correctly suggests, the carbon emissions caused by prediction errors may be influenced by many factors, e.g., startup and shutdown of thermal generators. We agree with you that this issue should be carefully evaluated via a refined simulation of power system operation, e.g., unit commitment. However, this is out of the scope of this paper that aims at the uncertainty analysis of renewable generation. Hence, we remove the discussions about carbon emissions caused by prediction error (mainly Fig.1 (c) and (d)) to make the paper more concentrated and solid. This revision will not impact the main findings and results. The evaluation of carbon emission caused by renewable generation uncertainty deserves future investigation, which can be simulated based on the analysis and dataset of this paper.

Action:

The contents related to uncertainty-induced carbon emissions have been removed. Instead, we elaborate on the impacts of different methods and time scales on the renewable prediction errors.

Comment 1.3

To my understanding, in practice system operators and planners tend to predict/forecast output conservatively. In this way, if the actual output is higher than the predicted one, the energy can be curtailed. This is due to the consideration of power system reliability. There are also many other factors affecting predict/forecast output, such as renewable resources' forecast accuracy.

I suggest the authors at least provide some discussions on this, or even analyze its impact. **Response:**

We appreciate the valuable comments and suggestions from this reviewer.

We agree with the reviewer that actual output of renewable generation might be curtailed by system operators, which will introduce additional deviation compared with the forecasted power output. However, this paper focuses on the inherent uncertainty of renewable generation, which is defined in this paper as the difference between the real generation potential and the forecasted result. Note that the generation potential is determined by weather conditions. In contrast, the actual realized renewable generation is impacted by power curtailment and power adjustment, which rely on factitious factors including operation instructions or market clearing.

In practice, forecasting and system scheduling are two separate processes for power system operation. Renewable forecasting is an important input of the system scheduling. In this paper, we focus on characterizing the spatiotemporal distributions of forecasting errors that reveal the inherent uncertainty of renewable generation. Hence, the preferences of operational strategies are not incorporated in the error analysis.

Action:

We have made a discussion in the Conclusion section, shown as follows:

“In this paper, we focus on the inherent uncertainty of renewable generation and the forecasting errors are obtained merely by time-series analysis. In practice, the prediction errors of renewable generation may be impacted by more complicated factors such as weather forecasting quality and operational curtailment strategies. In some application scenarios, the forecasting tools may result in asymmetric errors conservatively. For instance, a system operator tends to forecast renewable generation conservatively for the sake of system reliability. These practical factors may lead to deviations to the distribution of the forecasting error, and can be incorporated in the analysis by replacing the benchmark forecasting model with a more realistic one, which deserves an in-depth investigation in the future.”

Comment 1.4

Fig 1d provides carbon emissions and the proportion caused by prediction errors. Which is the source of China’s power sector’s CO₂ emissions and installed capacity of different generation technologies from 2021 to 2030 in Fig 1d? I think there are different trajectories and I suggest the authors provide a sensitive analysis in Fig 1d, or change this figure to other forms. By the way, in line 153 it is CO₂ emissions, not C02 emissions.

Response:

We appreciate the valuable comments and suggestions from this reviewer.

The source of the simulation results from 2021 to 2030 is from [R2]. In addition, as you suggest, the contents related to uncertainty-induced carbon emissions have been removed. Instead, we elaborate on the impacts of different methods and time scales on the renewable prediction errors.

[R2] Zhuo, Z., Du, E., Zhang, N., et al. Cost increase in the electricity supply to achieve carbon neutrality in China. *Nat Commun.* **13**, 3172 (2022).

Comment 1.5

Fig 2, is it wind daily (in Fig. 2c) or hourly (in caption) difference? I think it should be hourly. Is it solar daily (in caption) or hourly (in Fig. 2d) difference? It should be daily.

Response:

We appreciate the valuable comments and suggestions from this reviewer.

The answer is yes. For wind power, the first-order difference is defined hourly, i.e., the hour-to-hour difference of wind generation potential normalized by the capacity. This is because wind generation is more volatile than solar power and does not have a regular daily pattern. For solar power, the first-order difference is defined daily, i.e., the day-to-day difference of solar generation potential normalized by the capacity. This is because the solar generation has a regular daily pattern and the hourly first-order difference will also be similar in different days.

In addition, both hourly and daily first-order differences are examined for wind and solar generation. In Fig. r3c and Fig r3d, the most relevant indicators are presented in x-axis, while the other indicator is expressed by the bubble size in the plot. In Fig r3c, we can identify a significant linear correlation between prediction error and hourly first-order difference of wind generation. Bubble size in the figure illustrates the daily first-difference of wind generation, but it does not have a clear relationship with the wind prediction error. In Fig r3d, daily first-order difference is presented in x-axis. It can be seen that solar prediction has a strong linear relationship with the daily first-order difference. Bubble size indicates the hourly first-order difference, of which the relationship with the solar prediction error is not significant.

Fig. r3 Impacts of installed capacity, power generation and first-order difference of time series. a, wind installed capacity, b, solar installed capacity, c, wind hourly first-order difference, and d, solar daily first-order difference. Here we use daily and hourly data to analyze solar and wind energy, respectively, which are presented in the x-axis. Each bubble indicates less influential factors, including wind or solar generation, wind daily first-order difference,

and solar hourly first-order difference, respectively. The radius of each bubble is the value of each factor. The number of bubbles is 30, representing the 30 provinces of China, excluding Tibet (wind), Chongqing (solar), Hong Kong, Macao, and Taiwan. The black linear regression line fits the center of the bubbles. The color of each bubble indicates the different categories: red—category with the largest prediction error; yellow—category with the second-largest prediction error; blue—category with the third-largest prediction error; green—category with the smallest prediction error.

Action:

We have added a description in the revised manuscript:

“Due to the irregular distribution of the wind output and the daily periodicity of the solar output, we use hourly and daily output data to analyze the wind and solar prediction errors, respectively.”

Reviewer #2 (Remarks to the Author):

This is an impressive study on an important topic, which certainly merits publication. The authors examined the statistical characteristics of the inherent uncertainties of renewable generation arising from natural randomness. From a very fundamental and novel perspective, the numerical results reveal a pattern of renewable's spatiotemporal distributions across 30 provincial regions in China. Additionally, the authors try to answer a public interest as to how to quantify the extra carbon emissions induced by renewable uncertainty. Such a paradigm should work for similar knowledge discovery in other renewable-penetrated countries as well. The proposed method and database could potentially provide important references for stochastic-aware research and applications, which deserves a series of follow-up studies from different fields. In this instance, I suggest minor revisions as follows:

Comment 2.1

I was left a little unclear about the estimation for renewable uncertainty-induced carbon emissions. Can the authors attempt a wider context to explain the physical meaning of the estimation method? Also, how do the authors consider the uncertainty in the future?

Response:

We appreciate the valuable comments and suggestions from this reviewer.

In the original manuscript, we estimate the CO₂ emission caused by uncertainty of renewable generation. To balance the power supply and demand with uncertainty renewable generation, the power system needs to retain a certain level of reserve capacity. The reserve capacity can be provided by thermal generators, hydropower units and battery storages. In China, thermal generators are dominant resources of reserve capacity. The provision of reserve services will make thermal generators deviate from the most economic operating points, and thus will lead to additional fuel consumption and the consequent CO₂ emission. We use the difference between the maximum fuel consumption rate and the average fuel consumption rate in each province to estimate the uplift of thermal coal consumption for providing reserve service. Then the average CO₂ emission rate of thermal coal is multiplied to the results to estimate the additional CO₂ emission caused by renewable uncertainty.

As noted by Reviewer 1, we realize that the method for estimating CO₂ emission is coarse and there are many other factors that may impact the CO₂ emission. Since the focus of this paper is to characterize the inherent uncertainty of renewable generation rather than its emission effects, we remove the estimation of CO₂ emission from the revised manuscript.

Action:

The contents related to uncertainty-induced carbon emissions have been removed. Instead, we elaborate on the impacts of different methods and time scales on the renewable prediction errors.

Comment 2.2

It's interesting that the prediction error is highly dependent on the first-order difference and peaks of renewables. However, why do the authors define different calculation methods for wind and solar separately? In addition, the regression functions should be presented in Fig.2

and 3.

Response:

We appreciate the valuable comments and suggestions from this reviewer.

For wind power, the first-order difference is defined hourly, i.e., the hour-to-hour difference of wind generation potential normalized by the capacity. This is because wind generation is more volatile than solar power and does not have a regular daily pattern. For solar power, the first-order difference is defined daily, i.e., the day-to-day difference of solar generation potential normalized by the capacity. This is because the solar generation has a regular daily pattern and the hourly first-order difference will also be similar in different days.

In addition, both hourly and daily first-order differences are examined for wind and solar generation. In Fig. r4c and Fig r4d, the most relevant indicators are presented in x-axis, while the other indicator is expressed by the bubble size in the plot. In Fig r4c, we can identify a significant linear correlation between prediction error and hourly first-order difference of wind generation. Bubble size in the figure illustrates the daily first-difference of wind generation, but it does not have a clear relationship with the wind prediction error. In Fig r4d, daily first-order difference is presented in x-axis. It can be seen that solar prediction has a strong linear relationship with the daily first-order difference. Bubble size indicates the hourly first-order difference, of which the relationship with the solar prediction error is not significant.

Fig. r4 Impacts of installed capacity, power generation and first-order difference of time series. a, wind installed capacity, b, solar installed capacity, c, wind hourly first-order difference, and d, solar daily first-order difference. Here we use daily and hourly data to analyze solar and wind energy, respectively, which are presented in the x-axis. Each bubble indicates less influential factors, including wind or solar generation, wind daily first-order difference, and solar hourly first-order difference, respectively. The radius of each bubble is the value of each factor. The number of bubbles is 30, representing the 30 provinces of China, excluding Tibet (wind), Chongqing (solar), Hong Kong, Macao, and Taiwan. The black linear regression line fits the center of the bubbles. The color of each bubble indicates the different categories:

red—category with the largest prediction error; yellow—category with the second-largest prediction error; blue—category with the third-largest prediction error; green—category with the smallest prediction error.

Action:

We have added a description in the revised manuscript:

“Due to the irregular distribution of the wind output and the daily periodicity of the solar output, we use hourly and daily output data to analyze the wind and solar prediction errors, respectively.”

In addition, we have revised the figures by adding the regression functions.

Comment 2.3

Please further explain why the probability distribution functions of some intervals are absent in Fig.4. For example, Beijing has ten intervals but Inner Mongolia only has seven. Do the authors have any special consideration?

Response:

We appreciate the valuable comments and suggestions from this reviewer.

Fig. r5 Provincial probability distribution function (PDF) and prediction errors in each interval.

To analyze the distribution characteristics of prediction errors when wind/solar generates at different levels, we divide the total installed wind/solar generation capacity into 10 intervals for each province. In Fig. r5, the first and third rows exhibit the probability distribution functions (PDFs) of prediction errors when output falls into different intervals. The second and fourth rows exhibit prediction errors in different power output intervals in the form of box plots. In some provinces, the wind/solar generation is always lower than a certain proportion of the installed capacity.

In the revised manuscript, for instance, the solar generation in Inner Mongolia is always lower than 80% of its installed capacity. Thus, there is no value of solar generation and corresponding prediction errors in intervals of 80-90% and 90-100%. Hence, Inner Mongolia only has 8 intervals in the revised plot, while PDFs of prediction errors in the other 2 intervals are absent.

Action:

We elaborate on the reason of the absent intervals in the revised manuscript, shown as follows:

“To further explore the impact of different power generation levels on the prediction error, we evenly divide the installed generation capacity into 10 intervals.”

“PDFs and box plots are missing in some intervals because the power generation does not reach that range of the installed capacity, such as TJ wind generation only covers 0-90% capacity.”

Comment 2.4

Some other minor comments should include: i) The figures have to be further polished to comply with publication requirements. For example, the legends and curves in Fig.5 have overlaps. Also, it is ambiguous to justify the relationship between y-axis and the curve/bar. ii) Please further improve the quality of the policy implication part to demonstrate the role of the main contribution.

Response:

We appreciate the valuable comments and suggestions from this reviewer.

We have updated the figure to avoid overlap. We also justify the relationship between y-axis and the curve/bar in the caption of Fig. 5, shown as follows:

Fig. r6 Temporal analysis of wind and solar prediction errors. **a**, Wind, **c**, solar prediction error in the 30 provinces in spring, summer, autumn, and winter. Each chord and arc represent the prediction error (%) between a province and the season, where the thickness is proportional to the level of prediction error. Regarding province arcs, each segment corresponds to the prediction error in each season; regarding season arcs, each segment corresponds to the prediction error in each province. The number next to the arc indicates the cumulative prediction error. **b**, Hourly prediction error of wind power in Tianjin (TJ), Shandong (SD), Shanxi (SX), and Gansu (GS). **d**, Hourly prediction error of solar in Beijing (BJ), Jiangsu (JS), Hubei (HB), and Inner Mongolia (IM). Curves indicate hourly prediction errors (left axis), and bars indicate average prediction errors (right axis) in the four seasons: Green—spring; red—summer; yellow—autumn; and blue—winter.

Additionally, the policy implication has been highly improved:

“Based on the prediction error analysis, we summarize two policy suggestions for China. First, the government should provide adequate policy support and incentives to encourage wind energy development in the Southwestern and Central areas of China and solar energy development in the areas of Southwest and Northwest China. These areas experience limited fluctuations in wind and solar generation, around 2.1%-6.4% and 4.3%-7.4%, reducing the adverse impact on the power system. However, the current installed capacities in these regions are insufficient, even lower than East area with less land. Second, the government should plan interprovincial energy transmission in the space dimension to reduce the winter load peak in

North China and reduce the adverse impact of renewable energy. As concluded, the wind and solar fluctuations in North China are notable, accounting for 28.1% and 25.0%, respectively, of the total prediction error in China, especially during winter, with a proportion of 27.4% and 27.7%. However, during spring and summer, much energy consumption can be satisfied by renewable energy, resulting in an unbalance in different seasons and requiring additional energy sources. As such, the government should improve the power system infrastructure, systematically evaluate potential transmission projects, and plan additional power lines according to the resource and load distribution.”

REVIEWERS' COMMENTS

Reviewer #1 (Remarks to the Author):

The authors addressed most of my concerns. Minor revision is required.

The authors responded that the contents related to uncertainty-induced carbon emissions have been removed. I agree with the authors' decision. But in Page 16 the authors keep the methods to estimate carbon emissions. I suggest the authors remove this section. I have no other comments.

Reviewer #2 (Remarks to the Author):

Thanks for the effort addressing my comments. I suggest to accept this manuscript.

Response to Comments

Reviewer #1 (Remarks to the Author):

The authors addressed most of my concerns. Minor revision is required.

The authors responded that the contents related to uncertainty-induced carbon emissions have been removed. I agree with the authors' decision. But in Page 16 the authors keep the methods to estimate carbon emissions. I suggest the authors remove this section. I have no other comments.

Answer:

We appreciate the Reviewer's valuable comments and recognition of our work. We have removed methods for estimating carbon emissions on Page 16.

Reviewer #2:

Remarks to the Author:

Thanks for the effort addressing my comments. I suggest to accept this manuscript.

Answer:

The Reviewer's suggestions have been instrumental in improving our research work, and we are grateful for the effort in reviewing our paper.